# Autonomous error correction of a single logical qubit using two transmons

Ziqian Li [1,2,3,6], Tanay Roy [1,2,6], David Rodríguez Pérez[4], Kan-Heng Lee[1,2], Eliot Kapit[4] & David I. Schuster [1,2,3,5] ✉

Large-scale quantum computers will inevitably need quantum error correction to protect information against decoherence. Traditional error correction typically requires many qubits, along with high-efficiency error syndrome measurement and real-time feedback. Autonomous quantum error correction instead uses steady-state bath engineering to perform the correction in a hardware-efficient manner. In this work, we develop a new autonomous quantum error correction scheme that actively corrects single-photon loss and passively suppresses low-frequency dephasing, and we demonstrate an important experimental step towards its full implementation with transmons. Compared to uncorrected encoding, improvements are experimentally witnessed for the logical zero, one, and superposition states. Our results show the potential of implementing hardware-efficient autonomous quantum error correction to enhance the reliability of a transmon-based quantum information processor.

Quantum error correction (QEC) is critical for performing long computations involving many qubits, such as Shor's[1] or quantum chemistry algorithms[2]. Errors accumulating in the quantum system can be regarded as entropy or heat entering the system. In this context, the standard measurement and feedback-based QEC methods can be thought of us creating a "Maxwell Demon" keeping the system cold. These methods typically require many qubits and complex control hardware and have been demonstrated approaching the fault tolerance threshold[3–12]. When cooling atoms, rather than using measurement-based feedback, typically laser cooling is used. In laser cooling, the measurement and feedback are effectively encoded in the internal level structure and clever choice of laser drives. Along these lines, it is possible to perform autonomous quantum error correction (AQEC) where rather than measurements and gates, the system is "cooled" via an appropriate set of drives and couplings to engineered thermal reservoirs[13]. Like laser cooling, AQEC can dramatically simplify the quantum and classical hardware and control required. Both autonomous and feedback-based QEC are more challenging than simply cooling because they require that the cooling process preserves the logical manifold of the system. Here, we develop an AQEC code in a pure transmon-based[14] system using scalable on-chip circuit structures and experimentally demonstrate an important step towards its full implementation.

AQEC has received growing attention in theoretical proposals[15–23]. In addition to the usual QEC conditions[24], AQEC requires that the error-correction operations must commute with the system Hamiltonian at all times. This makes AQEC most appropriate for hardware efficient[25–30] systems with constrained error syndromes. Thus far all demonstrations, have encoded the logical qubits into 3D superconducting cavities using an ancilla qubit as a control[25,28].

In this work, we propose and implement a new AQEC protocol, called the Star code[31], that only requires easy-to-realize two-photon interactions. We develop a coherence-preserving two-transmon coupler that can parametrically generate all interactions needed for the protocol. With AQEC turned on, the logical states show higher coherence times than the uncorrected case limited by stray ZZ coupling between transmons. This is a crucial step towards fully demonstrating the Star code. The structure of the paper is as follows. First, we explain

[1]James Franck Institute, University of Chicago, Chicago, IL 60637, USA. [2]Department of Physics, University of Chicago, Chicago, IL 60637, USA. [3]Department of Applied Physics, Stanford University, Stanford, CA 94305, USA. [4]Department of Physics, Colorado School of Mines, Golden, CO 80401, USA. [5]Pritzker School of Molecular Engineering, University of Chicago, Chicago, IL 60637, USA. [6]These authors contributed equally: Ziqian Li, Tanay Roy.
✉e-mail: dschus@stanford.edu

the logical encoding and Hamiltonian construction of the Star code. Then we experimentally calibrate each of the parametric processes used in the code. Finally, we prepare the logical states and characterize the coherence improvement.

## Results

### Logical encoding

The Star code encodes a logical qubit using two orthogonal states in a nine-dimensional (two-qutrit) Hilbert space as $|L_0\rangle = (|gf\rangle - |fg\rangle)/\sqrt{2}$ (logical "zero"), and $|L_1\rangle = (|gg\rangle - |ff\rangle)/\sqrt{2}$ (logical "one") where $|g\rangle$, $|e\rangle$, and $|f\rangle$ represent the lowest three energy levels of a transmon. The error states after a single photon-loss (one transmon in $|e\rangle$) are orthogonal to the logical space and to each other. Further, both logical states have an equal expected photon number so that the single-photon loss (transmon decay) does not reveal information about the state it was emitted from. We engineer a parent Hamiltonian for the logical states through $|gf\rangle\langle fg|$ and $|gg\rangle\langle ff|$ parametric processes. These processes are all implemented by driving through $|ee\rangle$ as an intermediate state, producing the star topology in Hilbert space that gives the code its name (see Fig. 1a). Using an intermediate state allows these to be achieved using only 2-photon drives (QQ sidebands). We use the convention of calling them QQ red (single-photon exchange with low-frequency drives) and blue (two-photon pumping with high-frequency drives) sidebands. Despite both sets of drives going through $|ee\rangle$, the logical states can be made dark with respect to $|ee\rangle$ by detuning the $|L_0\rangle$ ($|L_1\rangle$) sidebands by $\pm \nu_r$ ($\pm \nu_b$) and setting equal drive strength $W$. When all of these processes are simultaneously applied, the two-transmon Hamiltonian in the logical-static frame (see

Supplementary Note 3 for derivation) is

$$\tilde{H}_{QQ} = \frac{W}{2} \left( |ee\rangle\langle gf| e^{2\pi i \nu_r t} + \|ee\rangle\langle fg| e^{2\pi i \nu_r t} \right. \\ \left. + |ee\rangle\langle gg| e^{2\pi i \nu_b t} + |ee\rangle\langle ff| e^{2\pi i \nu_b t} \right) + h.c. \quad (1)$$

Each transmon $Q_j$ is coupled to a lossy resonator $R_j$, which acts as the cold reservoir for dumping entropy. A single-photon loss, the dominant source of error in the system, populates the $|e\rangle$ level, triggering autonomous correction enabled by two transmon-resonator (QR) error correcting sidebands $|e0\rangle_j \leftrightarrow |f1\rangle_j$, $j = 1, 2$ (Fig. 1(b)). These sidebands are applied resonantly at rates $\Omega_j$ to the system, adding $\tilde{H}_{QRj}$ to the system Hamiltonian $H_{static}$,

$$\tilde{H}_{QR1} = \frac{\Omega_1}{2} a_{r1}^\dagger \left( |fg\rangle\langle eg| + |ff\rangle\langle ef| \right) \otimes I_4 + h.c., \quad (2)$$

$$\tilde{H}_{QR2} = \frac{\Omega_2}{2} a_{r2}^\dagger \left( |gf\rangle\langle ge| + |ff\rangle\langle fe| \right) \otimes I_4 + h.c., \quad (3)$$

$$\tilde{H}_{static} = \tilde{H}_{QQ} \otimes I_4 + \sum_{j=1,2} \tilde{H}_{QRj} + H_c. \quad (4)$$

Here $a_{rj}$ is the annihilation operator for the $j$-th resonator. $H_c$ contains the diagonal terms from frame transformation (See Supplementary Note 3 for full expression). We label the full state as $|Q_1 Q_2 R_1 R_2\rangle$ and keep the lowest two levels for each resonator. Here $I_n$ is the $n \times n$ identity matrix.

The Star code can correct the loss of a single photon from one of the qubits. Suppose $Q_1$ undergoes a photon-loss ($|f\rangle \to |e\rangle$) at rate $2\gamma_1$, where $\gamma_1$ is the $|e\rangle \to |g\rangle$ decay rate. The logical $|L_0 00\rangle$, consequently, becomes the error state $|eg00\rangle$ with energy $-\frac{\alpha_1}{2}$. Here $\alpha_j$ is the anharmonicity of $j$-th transmon. When $W \gg \Omega_1$, $\tilde{H}_{QR1}$ is a perturbation and only drives the transition $|eg00\rangle \leftrightarrow |L_0 10\rangle$ (See the left part of Fig. 1(b)). Assuming the resonator's decay rate $\kappa_1 \gg \gamma_1$, this oscillation quickly damps back to the original logical state $|L_0 00\rangle$ with no extra phase accumulated, and completes the correction cycle. The correction procedure for $|L_1\rangle$ is similar through an independent path (See the right part of Fig. 1(b)). The logical superposition state preserves relative phases since the QR sidebands do not distinguish the correction path. Such a two-step logical refilling rate can be approximated with Fermi's golden rule $\Gamma_{Rj} \simeq \frac{\Omega_j^2 \kappa_j}{\Omega_j^2 + \kappa_j^2} 32$. Apart from providing protection against single-photon loss, the star code also provides suppression to $1/f$ dephasing error[19,33] through spn-locking. The continuous QQ drives create an energy gap of $O(W)$ (right part of Fig. 1(a)) between the logical manifold and all other states, suppressing low-frequency noise. In the presence of only QQ drives (Fig. 1(a)), the "QQ echoed" qubit having the same logical manifold enjoys dephasing protection, and is a stepping stone towards the full code. Theoretical lifetime improvement of logical states is further discussed in Supplementary Note 3 and ref. 31.

We realize this protocol using the circuit shown in Supplementary Note 1. The key component is the inductive coupler based on the design in ref. 34 that enables the realization of fast parametric interactions. Two transmons $Q_1$ and $Q_2$ serve as the qutrits and share a common path to ground. This path is interrupted by a Superconducting Quantum Interference Device (SQUID) loop. The SQUID functions as a tunable inductor with external DC and RF magnetic fields threaded for activating the QQ sidebands. Each transmon is capacitively coupled to a lossy resonator serving both as the readout and cold reservoir. QR sidebands can be performed by sending a

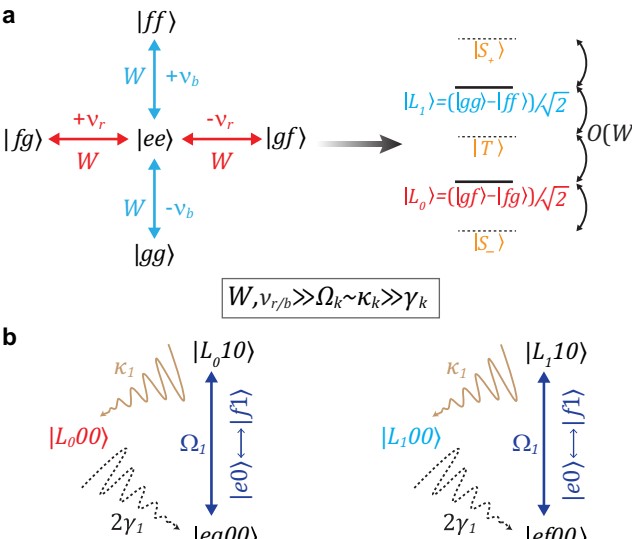

**Fig. 1 | Illustration of the autonomous error-correction scheme.** The protocol requires simultaneous application of two QQ blue sidebands ($|ee\rangle \leftrightarrow |gg\rangle$ and $|ee\rangle \leftrightarrow |ff\rangle$), two QQ red sidebands ($|ee\rangle \leftrightarrow |fg\rangle$ and $|ee\rangle \leftrightarrow |gf\rangle$), and two QR error correcting sidebands ($|e0\rangle \leftrightarrow |f1\rangle$). All six drives are always-on. **a** Star code logical word formation. All QQ sidebands have nominally equal rates $W$. The two drives within a pair have opposite detunings from the on-resonance values. This opens up the energy gaps of $O(W)$ between logical states and other states $\{|S_\pm\rangle, |T\rangle\}$ (see Supplementary Note 3 for full expression). With only QQ sidebands on, this forms the "QQ echoed" qubit sharing the same logical states as the star code. **b** The AQEC cycle for $|L_0\rangle$ (left) and $|L_1\rangle$ (right) when a single-photon-loss event occurs. Logical state $|L_0 00\rangle$ ($|L_1 00\rangle$) loses a photon from transmon $Q_1$ at a rate $2\gamma_1$ and becomes the error state $|eg00\rangle$ ($|ef00\rangle$). QR error correcting sidebands (applied on-resonance) bring the state at rate $\Omega_1$ to $|L_0 10\rangle$ ($|L_1 10\rangle$) with one photon populating $R_1$. $R_1$'s photon decays quickly (at a rate $\kappa_1$) and recovers the original logical state. AQEC cycle for $Q_2$'s photon loss event is similar.

charge drive at the half transition frequency to the transmon[35]. Full circuit quantization is shown in Supplementary Note 2.

### Device implementation and sideband calibration

We first characterize the individual qubits and realize the required sidebands to create and correct the logical states. We adjust the DC flux point to minimize the Cross-Kerr coupling between transmons which can dephase the logical superposition states (See Supplementary Note 4 for further discussion). The measured Cross-Kerr couplings are all lower than 320 kHz while maintaining Ramsey dephasing times

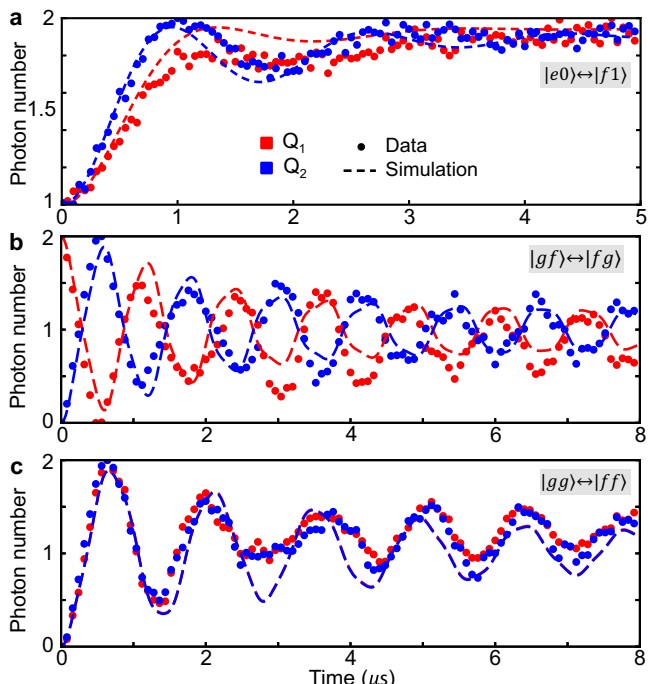

**Fig. 2 | Different parametric oscillations.** Photon numbers in individual transmons are measured as a function of time. **a** Error correcting QR sidebands $|e0\rangle \leftrightarrow |f1\rangle$ applied separately at rates $\Omega_1 = 0.49$ MHz and $\Omega_2 = 0.59$ MHz to the transmon-resonator pairs with $|e\rangle$ as initial states. Effective transitions **b** $|gf\rangle \leftrightarrow |fg\rangle$ and **c** $|gg\rangle \leftrightarrow |ff\rangle$ are measured when all QQ and QR sidebands are simultaneously turned on. Extracted sideband rates and detunings from simulation are $W_r = 1.45$ MHz, $W_b = 1.25$ MHz, $\nu_r = 0.8$ MHz, $\nu_b = -0.9$ MHz, $\Omega_1 = \Omega_2 = 0.39$ MHz. Other parameters are based on experimentally measured data shown in Supplementary Note 1 and 2. Oscillation distortions qualitatively match the lab frame simulations (dashed lines). Measurement errors are smaller than the marker size.

$T_{R_{ge}} = 15.2(9.8)\,\mu s$ with relaxation time $T_{1ge} = 24.3(9.1)\,\mu s$, for $Q_1(Q_2)$ (See Supplementary Note 1).

To calibrate the QR sidebands for selective photon pumping, we initialize the system in $|eg00\rangle$ and apply a continuous charge drive at frequency $(\omega_{r1} + \omega_{q1} + \alpha_1)/2$ to activate a 2-photon $|e0\rangle \leftrightarrow |f1\rangle$ transition between $Q_1$ and $R_1$ at a rate of 0.49 MHz. The system achieves a steady state $|fg00\rangle$ within 3 $\mu s$ as shown by red points in Fig. 2(a). Similarly, a 0.59 MHz QR2 drive takes $|ge00\rangle$ to $|gf00\rangle$ in a similar time (blue points in Fig. 2(a)). The decay of transmon reduces the final average photon number slightly below 2.

We achieve at least 20 MHz QQ red sidebands $(|j, k\rangle \leftrightarrow |j+1, k-1\rangle)$ and 5 MHz QQ blue sidebands $(|j, k\rangle \leftrightarrow |j+1, k+1\rangle)$ separately at the operating point, demonstrating a fast, coherence-preserved two-qutrit coupler with suppressed $ZZ$ interaction. Blue sidebands have a slower rate limited by stray signals from higher flux modulation frequencies (See discussion in Supplementary Note 5). All possible sidebands realized in this coupler are shown in Supplementary Note 6.

By driving all six sidebands, the core effective 4-photon processes, $|fg\rangle \leftrightarrow |gf\rangle$ and $|gg\rangle \leftrightarrow |ff\rangle$ and the error-correcting QR drives can be realized simultaneously. In practice, the QQ red and blue sideband rates ($W_r = 1.45$ MHz and $W_b = 1.25$ MHz) are slightly different. When applying all sidebands, we choose a smaller $W$, because the coupler was found to heat and shift the readout resonator when driven at larger rates making tomographic reconstruction inaccurate. We choose almost opposite detunings ($\nu_r = 0.8$ MHz, $\nu_b = -0.9$ MHz) for larger energy separation of the eigenstates and better error correction performance. Both QR sidebands are turned on at rates $\Omega_1 = \Omega_2 = 0.39$ MHz. Figure 2(b) shows the evolution when the initial state is $|fg\rangle$. The average photon number of $Q_1$ (in red) and $Q_2$ (in blue) are read out simultaneously, and the oscillation between 0 and 2 forms an effective 4-photon red sideband. Note that this effective swap process is slightly different from the direct $|fg\rangle \leftrightarrow |gf\rangle$ transition as the population in $|ee\rangle$ will appear intermediately when the initial state has overlap with the eigenstates that have $|ee\rangle$ component. Under this condition, $|ee\rangle$ is no longer the dark state when all six sidebands are on. Oscillation damping originates from the detuning-induced slow interference and decoherence of the qutrit subspace, and this distortion is captured by the simulation as well. Similarly, by choosing the initial state as $|gg\rangle$, the effective four-photon blue sideband $|gg\rangle \leftrightarrow |ff\rangle$ can be observed in Fig. 2(c).

### Error correction performance

The logical state initialization requires sequential application of multiple single-qutrit and two-qutrit rotations. For $|L_0\rangle$ and $|L_1\rangle$, QQ red and blue sidebands are used to generate entanglement, and for $|L_x\rangle = (|L_0\rangle + |L_1\rangle)/\sqrt{2} = (|g\rangle + |f\rangle)(|g\rangle - |f\rangle)/2$, only single qutrit

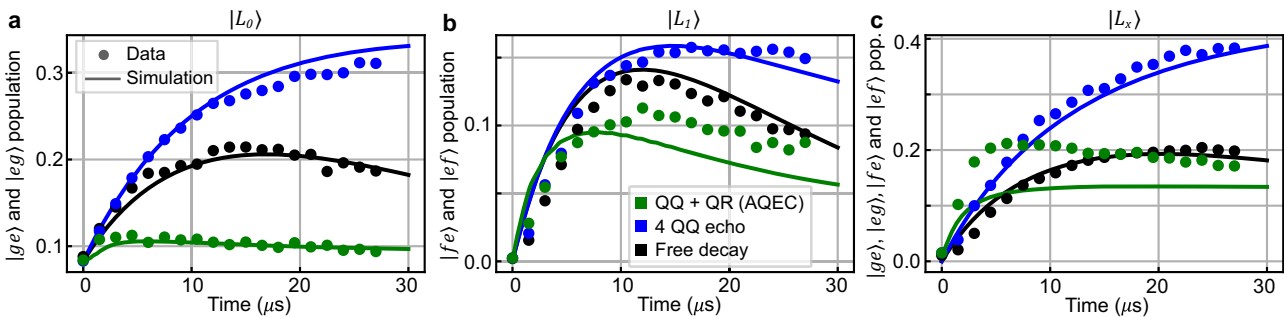

**Fig. 3 | Error population under different conditions.** Black, blue, and green points represent tomographic measurement results under free decay, 4 QQ echo, and full AQEC. The y-axes represent the combined population of error states for initial states **a** $|L_0\rangle$, **b** $|L_1\rangle$, and **c** $|L_x\rangle$. Population accumulates at the error states in the free decay case, enhanced in the 4 QQ echo case, and corrected with AQEC drive on. The experimental data is explained with master equation simulations shown in solid lines. Error bars (one standard deviation) are smaller than the marker size[38]. Detailed fitting parameters used for the solid lines are shown in Supplementary Note 10.

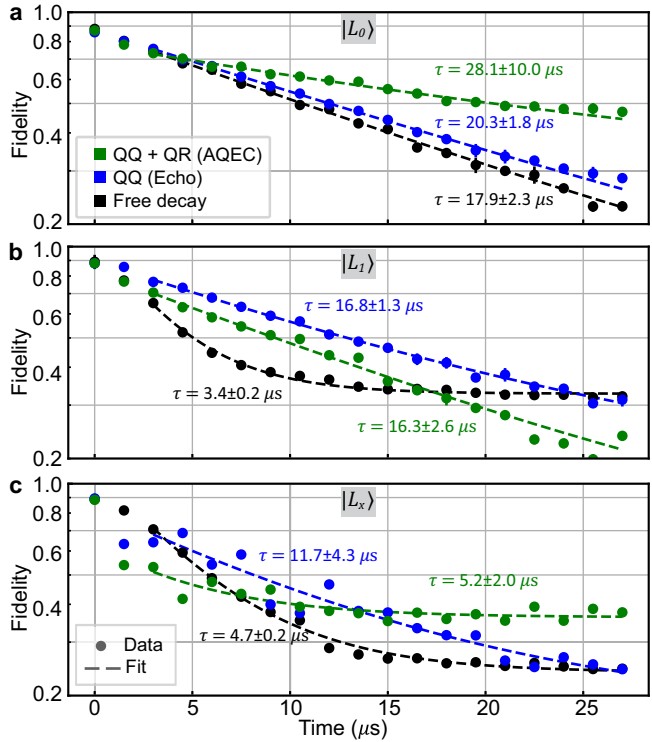

**Fig. 4 | Coherence improvement.** Black, blue, and green circles are experimentally measured state fidelities at a given time. The fidelities are extracted from tomographic reconstruction of states with 5000 repeated measurements. Error bars (one standard deviation) are smaller than the marker size[38]. The improvement with AQEC turned on is explained by the master equation simulation. **a** $|L_0\rangle$ and **b** $|L_1\rangle$ traces are fitted to the exponential decay curve $A\exp(-t/\tau)+C$, and **c** $|L_x\rangle$ traces are fitted to $A\exp(-t/\tau)$. The offset $C$ is necessary since fidelity will achieve steady-state values. The error (one standard deviation) for $\tau$ are obtained from the fitting. The large uncertainty comes from treating $C$ as a free variable in the fitting. The fast transition period (first $1.5\,\mu s \sim \Omega_j^{-1}$ in the AQEC case is not included in the fitting* for a better representation of logical coherence. *First $10.5\,\mu s$ data are used in $|L_1\rangle$'s free decay case for better fitting.

rotations are required. The preparation times for initial states are separately 313 ns, 142 ns, and 282 ns for $|L_0\rangle$, $|L_1\rangle$ and $|L_x\rangle$. The detailed preparation circuit is discussed in Supplementary Note 7. We perform full two-qutrit state tomography[36,37] and obtain initial state fidelities of 88.1%, 89.1% and 88.7% for the three states respectively. The tomography sequences and density matrix reconstruction are shown in Supplementary Note 8.

We characterize the performance of the Star code by comparing three different cases − free decay, QQ sideband spin-locking (4 QQ echo), and full AQEC. For free decay, we do not apply any drive after the state preparation. For the 4 QQ echo case, we turn on the QQ sidebands $|ee\rangle \leftrightarrow \{|gf\rangle, |fg\rangle, |gg\rangle, |ff\rangle\}$ with a similar rate-detuning configuration as shown in Fig. 1a ($W_r = 1.0$ MHz, $W_b = 1.7$ MHz, $\nu_r = 1.5$ MHz, $\nu_b = 0.0$ MHz). This case shows coherence improvement from spin-locking. The full AQEC ($W_r = 1.45$ MHz, $W_b = 1.25$ MHz, $\nu_r = 0.8$ MHz, $\nu_b = -0.9$ MHz, $\Omega_1 = \Omega_2 = 0.39$ MHz) demonstrates further improvement from photon-loss correction. We plot the density matrices of the logical states after preparation and after $9\,\mu s$ in Supplementary Note 8 for reference.

To demonstrate that our protocol corrects single-photon loss error, in Fig. 3, we plot the combined population of error states as a function of time for all three cases. The error populations are computed through the expectation values of $\varepsilon_0 = |ge\rangle\langle ge| + |eg\rangle\langle eg|$ for $|L_0\rangle$, $\varepsilon_1 = |ef\rangle\langle ef| + |fe\rangle\langle fe|$ for $|L_1\rangle$, and $\varepsilon_0 + \varepsilon_1$ for $|L_x\rangle$ corresponding to the states after single-photon loss. We extract the error population

from the density matrices reconstructed with full two-qutrit state tomography at each time point up to 27 $\mu s$ using the Maximum Likelihood Estimation (MLE) from 5000 repetitions of 81 different pre-rotation measurements for each state. This is a direct demonstration of the AQEC's effectiveness, as it measures the error state population designed to correct by the protocol. Compared to the free decay cases (black dots), turning on the AQEC clearly corrects photon loss and suppresses the error rate below the free decay cases (green dots). The error rates for all three logical states increase in the 4 QQ echo case (blue dots), as enhanced qutrit decay rates in the presence of sideband can lead to extra photon loss.

In addition to correcting photon loss, it is also important to characterize how well the AQEC protocol preserves the coherence of the logical states. To quantify the coherence, we plot the fidelity decay for each logical state. Fidelities are calculated as $F = (tr\sqrt{\sqrt{\rho_m}\rho_{th}\sqrt{\rho_m}})^2$, where $\rho_{th} = |L_i\rangle\langle L_i|$, $i = 0, 1, x$ depending on the compared logical states, and $\rho_m$ is the experimentally measured density matrix. All expectation values' error bars are calculated using the Tomographer package[38]. Fitting the data to the exponential decays for $|L_0\rangle$ and $|L_1\rangle$[39], the logical states' coherence are improved from 17.9 $\mu s$ ($|L_0\rangle$) and 3.4 $\mu s$ ($|L_1\rangle$) in the free decay cases, to 20.3 $\mu s$ and 16.8 $\mu s$ in the four QQ echo cases, and up to 28.1 $\mu s$ and 16.3 $\mu s$ in the error correction cases (see Fig. 4(a, b)). This demonstrates a factor of 1.6 and 4.8 improvement in logical state coherence against the free decay case. We believe the coherence limit of $|L_0\rangle$ and $|L_1\rangle$ is limited by QR sidebands not being sufficiently small compared to the QQ sidebands, which causes leakage into the $\{|T\rangle, |S_\pm\rangle\}$ states. $|L_x\rangle$'s lifetime after error correction is on par with the free decay case. The uncompensated ZZ coupling between transmons dephases the $|L_x\rangle$ after error correction, but not for $|L_0\rangle$ and $|L_1\rangle$. Therefore, we expect $|L_x\rangle$'s coherence time should be worse than that for $|L_0\rangle$ and $|L_1\rangle$. The fact that turning on QR sidebands does not significantly worsen the $|L_x\rangle$ coherence shows that without photon loss error, QR sidebands do not introduce significant dephasing error.

The large difference in free-decay coherence times between $|L_0\rangle$ and $|L_1\rangle$ originates from the low-frequency dephasing noise on $\Phi_{DC}$ through the flux line. It causes a shift in both transmons' frequencies in the same direction, which $|L_1\rangle$ is sensitive to but $|L_0\rangle$ is not. The passive echo protection from the Star code drives suppresses this; consequently, in the 4 QQ echo case both logical states have similar coherence time.

The AQEC performance is primarily limited by three factors in our experiment. The most important fact is that the QQ sideband rates $W_r$ and $W_b$ are well below their ideal values. Stronger drives would further suppress phase noise (lifetimes in the 4 QQ echo experiment are well below $2T_1$, indicating room for improvement), and the increased energy separation would also allow us to use stronger QR drives, correcting photon loss more quickly. Although the coupler supports 9 MHz QQ sidebands for short periods, when $W_b$ goes beyond 5 MHz the readout resonator frequency starts to shift, introducing systematic measurement distortion (See Supplementary Note 5 for details). This problem worsens with all six tones applied and we stay well below this limit to ensure reliable tomography results. The second limit is the ZZ coupling between the transmons, an extra dephasing channel for superposition states (see Supplementary Note 4 for details). Our coupler is operated at the minimum ZZ flux bias of the coupler to minimize the effect. It could be further mitigated by stronger QR sidebands enabling faster error correction, or through additional off-resonant QQ drive terms to dynamically cancel it. The third limit comes from heating and physical coherence drop when sidebands are turned on. The average photon number in the readout increases from < 0.01 (free decay and 4 QQ echo cases) to 0.03 (AQEC case), which will actively convert a logical state to the error state under the QR interactions and significantly reduces logical lifetime. Photon-excitation events in the transmon are also non-correctable errors, but

they should make smaller contributions to logical coherence (see Supplementary Note 10). Further improvement can thus come from two paths—improving isolation between control signals or improving physical qubit coherence so that weaker drives can be more effective. Other limits are in the order of milliseconds (see Supplementary Note 10) and do not affect our results considerably.

## Discussion

In summary, we have experimentally demonstrated all interactions required for a hardware-efficient AQEC code, the Star code, utilizing only two transmon-resonator pairs and a linear coupler to perform the second-order transitions. Three levels per transmon are used to store information, with the middle level capturing photon loss error, and entropy is dumped to the resonator autonomously through the always-on cooling sidebands. Inter-transmon parametric drives are applied to the coherence-preserving coupler for separating the Star code logical space from other eigenstates. We demonstrated a clear low-frequency dephasing suppression by turning on all QQ sidebands and a minor improvement after turning on the additional error correction drives because of the presence of residual $ZZ$ coupling. The static ZZ is suppressed with the inductive coupler while engineering the cancellation of the dynamical ZZ arising in the presence of all sidebands remains a topic of future research (see Supplementary Note 6 for details). Our system is entirely constructed from scalable components and fundamentally avoids the need for fast and accurate error detection and feedback error correction pulses. The Star code can be a self-corrected building block for the surface code[3,40] to further correct higher-order errors when scaled up, and can be a fault-tolerant qubit for the bosonic system.

In the future, engineering a $ZZ$-free coupler would remove the primary source of decoherence in this work. Error-transparent single-qubit and two-qubit gates have been proposed theoretically to extend the Star code beyond single qubits[17]. The Star code can also be implemented in other platforms with an anharmonic three-level structure.

## Methods

Error bars (one standard deviation) for all expectation values calculated from the Maximum Likelihood Estimation(MLE) reconstructed density matrix (See Supplementary Note 8) use the Tomographer package[38].

## Data availability

Source data are provided with the paper[41]. All data used within this paper are available from the corresponding author upon request.

## Code availability

Simulation codes are provided with the paper[41]. All other codes used within this paper are available from the corresponding author upon request.

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

## Acknowledgements

This work was supported by AFOSR Grant No. FA9550-19-1-0399 and ARO Grant No. W911NF-17-S0001. We thank Xinyuan You for useful discussions during the manuscript writing. Devices are fabricated in the Pritzker Nanofabrication Facility at the University of Chicago, which receives support from Soft and Hybrid Nanotechnology Experimental (SHyNE) Resource (NSF ECCS-1542205), a node of the National Science Foundation's National Nanotechnology Coordinated Infrastructure. This work also made use of the shared facilities at the University of Chicago Materials Research Science and Engineering Center, supported by the National Science Foundation under award number DMR-2011854. EK's research was additionally supported by NSF Grant No. PHY-1653820.

## Author contributions

E.K., and D.I.S. conceived the experiment. Z.L. designed the device, and T.R. fabricated the device using K.L.'s recipe. Z.L. calibrated the experiment and analyzed the data with assistance from T.R. Z.L. performed the simulation with help from T.R., D.P., and E.K. E.K. provided theoretical support and guidance throughout the experiment, and D.I.S. supervised all the aspects of the project. Z.L., T.R., and D.I.S. wrote the manuscript, with input from all the authors.

## Competing interests

The authors declare no competing interests.
