## [Peer Review File · Nature Communications]

Editorial Note: This manuscript has been previously reviewed at another journal that is not operating a transparent peer review scheme. This document only contains reviewer comments and rebuttal letters for versions considered at *Nature Communications*.REVIEWERS' COMMENTS

Reviewer #1 (Remarks to the Author):

I would like to thank the authors for satisfactorily addressing my comments from previous rounds and updating the claims made in reference 31. I can recommend the manuscript for publication in Nature Communications in its present form.

Reviewer #2 (Remarks to the Author):

I had recommended publication of the manuscript in Nature Physics after the authors properly weakened their claim: They should be clear that the QR sidebands implemented in the experiment could not yet achieve phase-coherent correction of photon loss errors.

The authors have now made this point clear. Indeed, “without photon loss error, QR sidebands do not introduce significant dephasing error” is the modest but valuable point to note here.

The authors have also made the update in presenting the state fidelity in 3x3 Hilbert space in Fig. 4, in a concession in response to another referee’s report. This represents a more standard metric to characterize the system performance. It leads to an explicit weakening of the original claim of the paper, which I also agree is the correct thing to do.

The authors have also taken steps to update Ref. 31 as the extended analysis of this present manuscript, hence the issue of dual claims of first proposal has been addressed. (The abstract in Ref.31's arXiv PDF file has been updated, but the authors should be aware that its abstract in arXiv’s webpage metadata remains to be updated.)

Once again, even with the weakened claims, considering the overall body of work on device and control engineering, and its role as the first proposal and experimental demonstration of anharmonic qutrit/qudit based AQEC schemes, I still tend to rate it favorably even on the level of Nature Physics. However, I agree that the outcome of the experiments remains much to be desired. I support its publication in Nature Communications.

Reply to Referees

We sincerely thank the Referees for their assessment and constructive remarks and the Editor for showing interest in transferring this paper.

Below we address the comments of the Referees point-by-point and indicate the corresponding changes to the manuscript. We believe these remarks have contributed significantly to improving the quality of the manuscript. In addition, we have made some minor changes, and all changes in the main text and supplement are highlighted in blue. We hope you will consider this revised manuscript for publication in Nature Communications.

Sincerely yours,

Ziqian Li, Tanay Roy, Eliot Kapit, and David Schuster

REFEREE A

General comment: I would like to thank the authors for satisfactorily addressing my comments from previous rounds and updating the claims made in reference 31. I can recommend the manuscript for publication in Nature Communications in its present form.

Response: We thank the referee for recommending publication in Nature Communications.

REFEREE B

General comment: I had recommended publication of the manuscript in Nature Physics after the authors properly weaken their claim: They should be clear that the QR sidebands implemented in the experiment could not yet achieve phase-coherent correction of photon loss errors. The authors have now made this point clear. Indeed, “without photon loss error, QR sidebands do not introduce significant dephasing error” is the modest but valuable point to note here.

The authors have also made the update in presenting the state fidelity in 3x3 Hilbert space in Fig. 4, in a concession in response to another referee’s report. This represents a more standard metric to characterize the system performance. It leads to an explicit weakening of the original claim of the paper, which I also agree is the correct thing to do.

The authors have also taken steps to update Ref. 31 as the extended analysis of this present manuscript, hence the issue of dual claims of first proposal has been addressed. (The abstract in Ref.31’s arXiv PDF file has been updated, but the authors should be aware that its abstract in arXiv’s webpage metadata remains to be updated.)

Once again, even with the weakened claims, considering the overall body of work on device and control engineering, and its role as the first proposal and experimental demonstration of anharmonic qutrit/qudit based AQEC schemes, I still tend to rate it favorably even on the level of Nature Physics. However, I agree that the outcome of the experiments remains much to be desired. I support its publication in Nature Communications.

Response: We appreciate the referee’s high assessment of our work after the suggested modifications and thank the referee for catching the outdated arXiv abstract, which we has now been updated.